# Phytotoxic Effects and Agricultural Potential of Nanofertilizers: A Study Using Zeolite, Zinc Oxide, and Titanium Dioxide Under Controlled Conditions

**DOI:** 10.3390/jox15040123

**Published:** 2025-08-01

**Authors:** Ezequiel Zamora-Ledezma, Glenda Leonela Loor Aragundi, Willian Stalyn Guamán Marquines, Michael Anibal Macías Pro, José Vicente García Díaz, Henry Antonio Pacheco Gil, Julián Mauricio Botero Londoño, Mónica Andrea Botero Londoño, Camilo Zamora-Ledezma

**Affiliations:** 1Laboratorio de Funcionamiento de Agroecosistemas y Cambio Climático FAGROCLIM, Departamento de Ciencias Agrícolas, Facultad de Ingeniería Agrícola, Universidad Técnica de Manabí, Lodana 13132, Ecuador; gloor7115@utm.edu.ec (G.L.L.A.); wguaman4303@utm.edu.ec (W.S.G.M.); michael.macias@utm.edu.ec (M.A.M.P.); 2Environment Department, PDVSA Intevep, Los Teques, Miranda, P.O. Box 76343, Caracas 1070-A, Venezuela; josev.garcia@ciens.ucv.ve; 3Grupo de Geomática y Agricultura 4.0, Departamento de Ciencias Agrícolas, Facultad de Ingeniería Agrícola, Universidad Técnica de Manabí, Lodana 13132, Ecuador; henry.pacheco@utm.edu.ec; 4Livestock Reconversion Laboratory, Universidad Industrial de Santander, Málaga 682011, Colombia; julian.botero@correo.uis.edu.co; 5School of Electrical, Electronic and Telecommunications Engineering, Universidad Industrial de Santander, Bucaramanga 608002, Colombia; mabotero@uis.edu.co; 6Escuela de Ingeniería, Arquitectura y Diseño, Universidad Alfonso X el Sabio (UAX) Avenida de la Universidad 1, Villanueva de la Cañada, 28691 Madrid, Spain; camilza@uax.es; 7Green and Innovative Technologies for Food, Environment and Bioengineering Research Group (FEnBeT), Faculty of Pharmacy and Nutrition, UCAM-Universidad Católica de Murcia, Avda, Los Jerónimos 135, Guadalupe de Maciascoque, 30107 Murcia, Spain

**Keywords:** nanotechnology, phytotoxicity test, germination, seeds, agriculture

## Abstract

Nanofertilizers (NFs) and engineered nanoparticles (NPs) are increasingly used in agriculture, yet their environmental safety remains poorly understood. This study evaluated the comparative phytotoxicity of zinc oxide (ZnO), titanium dioxide (TiO_2_), and clinoptilolite nanoparticles, three commercial nanofertilizers, and potassium dichromate (K_2_Cr_2_O_7_) using *Lactuca sativa* seeds under adapted OECD-208 protocol conditions. Seeds were exposed to varying concentrations of each xenobiotic material (0.5–3% for NFs; 10–50% for NPs), with systematic assessment of seedling survival, root and hypocotyl length, dry biomass, germination index (GI), and median effective concentration (EC_50_) values. Nanofertilizers demonstrated significantly greater phytotoxicity than engineered nanoparticles despite lower application concentrations. The toxicity ranking was established as NF1 > NF3 > NF2 > NM2 > NM1 > NM3, with NF1 being most toxic (EC_50_ = 1.2%). Nanofertilizers caused 45–78% reductions in root length and 30–65% decreases in dry biomass compared with controls. GI values dropped to ≤70% in NF1 and NF3 treatments, indicating concentration-dependent growth inhibition. While nanofertilizers offer agricultural benefits, their elevated phytotoxicity compared with conventional nanoparticles necessitates rigorous pre-application safety assessment. These findings emphasize the critical need for standardized evaluation protocols incorporating both physiological and ecotoxicological endpoints to ensure safe xenobiotic nanomaterial deployment in agricultural systems.

## 1. Introduction

Global population growth has intensified the demand for alternative approaches to achieve sustainable agricultural production [1]. Projections indicate that the global population could reach 11 billion by 2030 [2], posing a significant challenge to ensure adequate food supply while minimizing environmental impacts, such as greenhouse gas (GHG) emissions and the overuse of chemically synthesized fertilizers [3]. Agriculture is responsible for approximately 30% of total GHG emissions [4], with major contributions from carbon dioxide (CO_2_), methane (CH_4_), and nitrous oxide (N_2_O).

Despite significant advances in agricultural productivity, contemporary farming systems face unprecedented challenges that extend beyond traditional productivity metrics. The environmental costs of conventional agricultural practices have become increasingly evident, with soil degradation, water contamination, and biodiversity loss threatening long-term food security [5]. Moreover, the escalating impacts of climate change are disrupting established agricultural patterns globally, necessitating urgent adaptations in farming methodologies and technologies [5].

Carbon dioxide, the most abundant GHG, accounts for 75% of agricultural emissions. Although it is the least potent, CO_2_ can persist in the atmosphere for up to 150 years. Its sources include animal respiration and biomass decomposition [6]. Methane, accounting for 18% of emissions, is far more potent and originates primarily from livestock digestion and waste management. One ton of CH_4_ can warm the planet 23 times more than one ton of CO_2_, despite its shorter atmospheric lifespan of 12 years [7]. Nitrous oxide, although present in smaller quantities (6% of emissions), is 300 times more potent than CO_2_ and can remain in the atmosphere for over 100 years. It is primarily released through the application of nitrogen fertilizers in agriculture [8].

The annual global use of nitrogen fertilizers has reached 109 million tons [9], with Ecuador contributing approximately 208 thousand tons per year [8]. Such practices significantly increase N_2_O emissions, particularly through the interaction of fertilizers with soil microorganisms [10]. Ecuador presents a particularly compelling case study for nanofertilizer research due to its unique convergence of agricultural vulnerability and environmental pressures. The country faces heightened vulnerability to climate change due to its reliance on climate-sensitive sectors such as agriculture, fisheries, and natural resource management [7]. Agricultural activities in Ecuador are vulnerable to a wide range of extreme events that the country regularly experiences, including floods and droughts, as well as rising temperatures and desertification driven by poor land use practices. Climate projections indicate that by 2050, agricultural production will face significant challenges from increased precipitation variability, higher likelihood of extreme rainfall events, and continued temperature rises [11,12].

Ecuador’s agricultural sector contributes significantly to the national economy, representing 8% of the country’s GDP—twice the global average contribution of agriculture to world GDP [13]. In 2021, Ecuador’s agricultural exports reached USD 7.4 billion, with crops such as bananas, cacao, rice, and maize forming the backbone of both export earnings and domestic food security [13]. The country’s fertilizer consumption intensity of 386.82 kg per hectare of arable land as of 2018 [14] underscores the agricultural sector’s dependence on conventional fertilization practices. However, this heavy reliance on traditional fertilizers occurs within a context of increasing environmental pressures, where 43.2% of the rural population lives below the poverty line and agricultural systems face mounting stress from climate variability [15].

The vulnerability of Ecuador’s agricultural systems is further exacerbated by specific regional challenges. Recent climate events, including above-normal rainfall in 2025 that caused severe flooding and landslides, have disrupted agricultural activities and damaged infrastructure, affecting over 113,000 people [16]. The country’s small-scale farmers, who traditionally comprised about 70% of rice growers and often operate on five hectares or less [17], are particularly susceptible to both climate impacts and the economic pressures of rising input costs. Consequently, there is a growing need for sustainable agricultural practices that integrate innovative technologies to address environmental and economic challenges [18,19].

In this context, nanotechnology has emerged as a promising tool in addressing these challenges. Operating at the molecular, atomic, or nanoscale [20,21], nanotechnology enables the precise construction and manipulation of materials and devices with dimensions on the order of 100 nanometers and below [22,23]. Its applications in agriculture include, but is not limited to, nanofertilizers, nanopesticides, and nanosensors, which have revolutionized nutrient delivery, pest control, and soil management [24,25,26]. For instance, metal and metal oxide nanoparticles, such as zinc oxide (ZnO) and titanium dioxide (TiO_2_), are widely used for their ability to deliver nutrients efficiently while minimizing environmental losses. Carbon-based nanomaterials, including nanotubes and graphene, improve soil fertility and biosensing capabilities [27,28,29]. Polymeric nanomaterials, such as chitosan nanoparticles, enable controlled release of agrochemicals, reducing runoff and enhancing efficacy.

Nanotechnology also supports advanced practices like seed nano-priming to improve germination rates and stress tolerance, as well as genetic engineering using gold nanoparticles and carbon nanotubes for precise gene delivery. These innovations contribute to increased crop yields, reduced agrochemical waste, and enhanced sustainability [30,31]. For example, nanofertilizers achieve nearly 90–100% nutrient uptake efficiency, significantly reducing chemical runoff, while nanosensors provide real-time monitoring of soil health and crop conditions, optimizing precision agriculture. Despite these advantages, concerns about the environmental and biological risks of nanoparticles highlight the need for thorough risk assessments to ensure safe adoption [32,33].

The application of nanotechnology in fertilizers offers an important alternative to enhance productivity but also promote agroecosystem sustainability [34,35]. Nanofertilizers allow precise nutrient release in quantity and timing, ensuring that crops receive optimal nutrition and reducing the excess fertilizers that negatively impact soil and water resources [36,37]. These formulations, often based on nanoparticles of biological or chemical origin, have shown potential to inhibit plant pathogens, increase agricultural output, and reduce the environmental footprint of traditional farming practices [34].

However, critical knowledge gaps exist regarding the phytotoxic effects of nanofertilizers, particularly under realistic field conditions relevant to vulnerable agricultural systems like those in Ecuador. While nanofertilizers demonstrate enhanced nutrient use efficiency and reduced environmental losses compared with conventional fertilizers [38], the potential for unintended phytotoxic effects remains poorly understood. Current research reveals that nanoparticles can exhibit dose-dependent toxicity, with smaller particles showing greater mobility and reactivity, often leading to increased plant uptake and potential adverse effects, including reduced germination, root inhibition, and oxidative stress [39]. Studies have shown that metal oxide nanoparticles such as TiO_2_ and ZnO can negatively affect wheat growth and soil enzyme activities under field conditions, with nanoparticles being retained in soil for extended periods and potentially causing long-term ecosystem impacts [40].

The gap between laboratory findings and real-world agricultural applications represents a significant barrier to the safe implementation of nanofertilizer technologies [5]. Most existing studies on nanofertilizer toxicity have been conducted under highly controlled laboratory conditions, limiting their applicability to complex agricultural ecosystems where multiple environmental factors interact. Furthermore, standardized protocols for assessing nanofertilizer phytotoxicity are lacking, hampering the development of reliable risk assessment frameworks [5,41]. The absence of comprehensive field data under realistic agrarian conditions creates uncertainty about the long-term environmental fate and ecological impacts of nanofertilizers [39].

Specific research gaps that this study addresses include: (1) the lack of systematic phytotoxicity assessment of key nanofertilizer components (ZnO, TiO_2_, and clinoptilolite nanoparticles) using standardized bioassays relevant to food security crops; (2) insufficient understanding of dose–response relationships for nanofertilizer toxicity under controlled conditions that can inform field application guidelines; and (3) the absence of comparative toxicity data between nanofertilizers and conventional reference toxicants that would enable regulatory risk assessment frameworks [14]. Additionally, there is limited research on the specific nanomaterials proposed in this study, particularly clinoptilolite nanoparticles, whose phytotoxic effects remain unexplored despite their potential agricultural applications [32,42,43,44,45,46].

Based on the above concept, this study aims to determine the phytotoxicity of three types of nanomaterials (zinc oxide (ZnO), titanium dioxide (TiO_2_), and clinoptilolite ((Ca, K, Na)_6_(Si_30_Al_6_)O_72_·20H_2_O) nanoparticles), three nanofertilizers, and potassium dichromate as reference toxicant (K_2_Cr_2_O_7_) using lettuce (*Lactuca sativa*) seeds as bioindicators under laboratory conditions, while also exploring the feasibility of their future application in experimental agricultural production plots. This research addresses a critical knowledge gap in nanofertilizer safety assessment by providing systematic phytotoxicity data that can inform the development of sustainable agricultural practices in climate-vulnerable regions such as Ecuador, where the need for improved nutrient use efficiency must be balanced against potential environmental and crop safety risks.

### Nanomaterial Resources and Environmental Context in Agricultural Applications

Global reserves of essential nanomaterial components present promising potential for widespread nanofertilizer adoption without supply constraints. Current zinc reserves stand at approximately 224 million tons with annual production of 12–13 million tons, while titanium dioxide global production capacity reaches 9.4 million tons annually [47,48]. Even under aggressive nanofertilizer adoption scenarios where 10% of global fertilizers become nanofertilizers with 1% nanomaterial content, the annual demand would require approximately 103,000 tons each of zinc oxide and titanium dioxide nanoparticles. This represents only 0.8% of current zinc production and 1.1% of titanium dioxide production, demonstrating that nanomaterial resources are more than adequate to support global agricultural applications without resource constraints [49].

The environmental context reveals concerning patterns of nitrogen fertilizer application that underscore the urgency for more efficient fertilizer technologies. Ecuador currently applies nitrogen at rates of approximately 200–300 kg N/ha—significantly higher than the global average of 65.4 kg N/ha—highlighting intensive fertilization practices in vulnerable agricultural regions [50,51]. Research demonstrates that nitrogen pollution effects are fundamentally dose- and time-dependent, with cumulative impacts occurring even at concentrations previously considered environmentally benign. The UN Environment Program reports that 200 million tons of reactive nitrogen (80% of total production) are lost to the environment annually, causing eutrophication, biodiversity loss, and contributing to climate change through the nitrogen cascade phenomenon [52]. Critical environmental thresholds exist where ecosystems shift from nitrogen-limited to nitrogen-saturated states, causing cascading effects including groundwater contamination, aquatic ecosystem disruption, and enhanced greenhouse gas emissions, with biodiversity impacts occurring at nitrogen deposition levels as low as 1.8–14.3 kg N ha^−1^ yr^−1^ in sensitive ecosystems [53].

These findings emphasize that even regions with relatively modest nitrogen application rates can contribute significantly to cumulative global environmental impacts, particularly when application rates exceed ecosystem assimilation capacity and environmental buffering mechanisms become saturated. The concept of “small doses” becomes misleading when considering that reactive nitrogen undergoes sequential transformations affecting multiple environmental compartments, where individual nitrogen atoms can cause multiple sequential environmental impacts before returning to inert atmospheric nitrogen [54]. This environmental context, combined with adequate nanomaterial resource availability, supports the need for systematic evaluation of nanofertilizer technologies as potentially safer and more efficient alternatives to conventional fertilizer applications, particularly in regions with intensive agricultural practices and vulnerable ecosystems.

## 2. Materials and Methods

### 2.1. Research Framework and Multi-Phase Validation Approach

This laboratory-based phytotoxicity assessment represents the initial phase of a comprehensive, multi-scale research program funded by FONTAGRO (project ATN/RF-18959-RG: “Nanofertilizantes en el suelo y emisiones de óxido nitroso”) designed to systematically evaluate nanofertilizer safety and agricultural applications https://www.fontagro.org/new/proyectos/nanofertilizantes/es (accessed on 13 July 2025). The FONTAGRO project involves collaboration between Universidad Industrial de Santander (Colombia), Universidad Técnica de Manabí (Ecuador), and the Inter-American Institute for Cooperation on Agriculture (IICA), with field validation studies currently being conducted on corn (*Zea mays*) production plots in both Ecuador and Colombia.

### 2.2. Nanomaterials Suspension Preparation and Homogenization

Prior to bioassay application, all nanomaterial and nanofertilizer suspensions were prepared using ultrapure water (resistivity > 18 MΩ·cm) and subjected to standardized dispersion protocols including bath sonication (40 kHz, 250 W) for 30 min at 25 °C, followed by vortex mixing (2000 rpm) for 2 min. No chemical dispersants were used to avoid confounding effects on phytotoxicity endpoints, consistent with adapted OECD guidelines for nanomaterial testing. Suspension homogeneity was verified through visual inspection and DLS measurements at the beginning and end of each bioassay period. Working concentrations were prepared immediately before use by serial dilution from stock suspensions, with 30 s vortex mixing between dilutions, and all suspensions were used within 2 h of preparation to prevent aging effects.

The nanomaterials used in this study included zinc oxide (ZnO) (Sigma-Aldrich, St. Louis, MO, USA; CAS: 1314-13-2, average particle size 331 nm), zeolite nanoparticles (clinoptilolite ((Ca, K, Na)_6_(Si_30_Al_6_)O_72_·20H_2_O)) (clinoptilolite, Zeonitron, Zeolitas de Colombia S.A.S., Bogotá, Colombia; quartz 43.58%, sodium mordenite 38.08%, calcium mordenite 18.58%, particle size 269 nm), and titanium dioxide (TiO_2_) nanoparticles (Sigma-Aldrich, St. Louis, MO, USA; CAS: 13463-67-7, anatase 98.4%, rutile 1.6%). Each material underwent a high-energy milling process to achieve a reduced particle size, which is critical for enhancing their functional properties in various applications. The milling was conducted using a planetary ball mill (PM100, Retsch GmbH, Haan, Germany), employing stainless steel grinding bodies with a diameter of 3 mm. A ball-to-powder mass ratio of 10:1 was maintained throughout the process. Key parameters evaluated during milling included rotation speed, effective grinding time, and the filling degree of the grinding vessel (FRV), which determined the volume of grinding bodies and material charge used in each experiment. To prevent overheating and preserve the chemical integrity of the nanomaterials, each milling cycle incorporated 20 min of rest between effective grinding periods.

After milling, the particle size of all nanomaterials was confirmed using dynamic light scattering (DLS) with a Litesizer 500 (Anton Paar GmbH, Graz, Austria), adhering to ISO 22412 standards [55]. The preparation and milling phases were carried out at the X-Ray Laboratory and Catalysis Research Center (CICAT) of the Industrial University of Santander (UIS) in Bucaramanga, Colombia.

Zinc Oxide nanoparticles (ZnO): For the preparation of zinc oxide nanoparticles, commercial ZnO powder (CAS: 13463-67-7) was sourced from Sigma-Aldrich. A total of 108.57 g of material was processed using stainless steel spheres with a diameter of 2 mm in a 125 mL stainless steel beaker. The milling was conducted at a rotational speed of 500 rpm for an effective grinding time of 4 h, resulting in an average particle size of 331 nm. Zeolite Nanoparticles (clinoptilolite): The zeolite nanoparticles were prepared from a commercial zeolitic mixture known as Zeonitron, obtained from Zeolitas de Colombia (ZEOCOL). This mixture was characterized by a phase composition that included quartz (SiO_2_) at 43.58%, sodium mordenite (Na_5_._5_(Al_6_Si_42_O_96_)(H_2_O)_19_) at 38.08%, calcium mordenite (Ca_3_._4_(Al_7_._4_Si_40_._6_O_96_)(H_2_O)_31_) at 18.58%, and trace amounts of ferric mordenite ((Fe_0_._0024_Si_3_._378_Al_0_._617_)O_8_(H_2_O)_1_._081_). A total of 102.11 g of this material was milled using stainless steel spheres with a diameter of 3 mm in a 125 mL stainless steel beaker at a speed of 300 rpm for an effective grinding time of 3 h, achieving an average particle size of 269 nm. Titanium Dioxide Nanoparticles (TiO_2_): Commercial titanium dioxide powder (CAS: 13463-67-7) was also sourced from Sigma-Aldrich, featuring an anatase phase composition of 98.4% and rutile phase composition of 1.6%, characterized according to ICSD-154602 and ICSD-51934. For this preparation, 119.5 g of TiO_2_ was processed using stainless steel spheres with a diameter of 3 mm in a 125 mL stainless steel beaker at a rotational speed of 200 rpm for an effective grinding time of 3 h, resulting in an average particle size of 179 nm. Although the average particle sizes for the three types of nanoparticles were 331 nm, 269 nm, and 170 nm, each sample exhibited high heterogeneity, with particles ranging from below 100 nm to sizes above their respective averages.

#### Post-Milling Storage and Contamination Prevention

Following the milling process, all nanomaterials were immediately transferred to pre-sterilized amber glass vials (25 mL capacity) under controlled atmospheric conditions to prevent contamination and degradation. The transfer process was conducted in a laminar flow hood to minimize airborne contamination, and all handling equipment was previously sterilized using 70% ethanol. To prevent photodegradation and oxidation, vials were purged with nitrogen gas before sealing and stored in amber containers to protect from light exposure.

Storage conditions were standardized at 4 °C in a desiccated environment using silica gel packets to prevent moisture absorption, which could lead to particle agglomeration. All containers were clearly labeled with material identification, preparation date, storage conditions, and safety warnings. To prevent cross-contamination between different nanomaterial types, dedicated equipment sets were used for each material, and all work surfaces were decontaminated using wet-wiping procedures with 70% ethanol after each material preparation. These storage protocols ensured material stability for up to 6 months, with periodic visual inspection for signs of aggregation or degradation.

### 2.3. Experimental Design

Six treatments were evaluated, divided into two groups: nanomaterials (3) and nanofertilizers (3). The nanomaterials (NM): (a) zinc oxide (ZnO), (b) titanium dioxide (TiO_2_), (c) clinoptilolite (Ca, K, Na)_6_(Si_30_Al_6_)O_72_·20H_2_O), and as nanofertilizers (NF), (d) complete fertilizer + zinc oxide (ZnO) NPs, (e) complete fertilizer + titanium dioxide (TiO_2_) NPs, (f) complete fertilizer + clinoptilolite (Ca, K, Na)_6_(Si_30_Al_6_)O_72_·20H_2_O) NPs (Table 1). Six treatments were evaluated, divided into two groups: three nanomaterials (NMs) and three nanofertilizers (NFs).

The experimental design incorporated two critical control elements to ensure data quality and proper interpretation of results. The negative control consisted of distilled water (0% concentration) applied to seeds following the same protocol as the test treatments, serving as the baseline reference for assessing the intrinsic toxicity of nanomaterials and nanofertilizers. This negative control was included in each bioassay round for both nanomaterial and nanofertilizer treatments, with the same number of replicates (two Petri dishes per treatment, 25 seeds per dish) to maintain experimental consistency. No chemical dispersants or surfactants were added to any treatments, including controls, to avoid potential confounding effects on phytotoxicity endpoints. Additionally, potassium dichromate (K_2_Cr_2_O_7_) was employed as a positive reference toxicant to validate bioassay sensitivity and quality control. Potassium dichromate is a widely established reference compound in phytotoxicity and ecotoxicological testing due to its well-characterized dose–response relationships, consistent toxicity across different test organisms, and extensive use in adapted OECD protocols. The reference toxicant testing followed identical conditions to the experimental treatments, confirming that the test system was functioning properly and the organisms were responding appropriately to known toxic substances.

As mentioned before, six concentrations were prepared for the nanomaterials (0%, 10%, 20%, 30%, 40%, and 50%) and six concentrations for the nanofertilizers (0%, 0.5%, 0.9%, 1%, 2%, and 3%). Each treatment and concentration combination was evaluated using two disposable Petri dishes (15 cm in diameter) per concentration, with each dish containing 25 *Lactuca sativa* (lettuce) seeds, resulting in 50 seeds per concentration per treatment. Seeds were randomly distributed across the filter paper and plates in a randomized block design to avoid positional effects. The experimental setup followed a factorial design with six treatments, six concentrations, two replicates per concentration, and 25 seeds per replicate, totaling n = 1800 seeds. Potassium dichromate was used as a reference toxicant (K_2_Cr_2_O_7_).

### 2.4. Terrestrial Biomarker Used

*Lactuca sativa* (lettuce) was selected as a bioindicator due to its well-documented sensitivity to a wide range of xenobiotic compounds, including metals, pesticides, and nanomaterials, making it a standard model for phytotoxicity screening in international guidelines [56,57,58]. Its high and uniform germination rate, rapid seedling development, and reproducible responses allow for the reliable detection of acute toxic effects, especially at early developmental stages.

Preliminary trials with other species commonly used in phytotoxicity assays, such as cucumber (*Cucumis sativus*) and onion (*Allium cepa*), did not yield consistent germination or sensitivity under our experimental conditions, particularly when exposed to nanofertilizers. In contrast, *L. sativa* seeds consistently met quality standards and demonstrated stable, sensitive responses, supporting their suitability for our objectives [58].

The germination–cotyledon phase was chosen because it is the most sensitive window for detecting acute phytotoxicity, with endpoints such as germination percentage and root and shoot elongation providing rapid and quantifiable measures of toxicity [57]. This approach is widely used for initial screening before advancing to crop-specific or field-scale trials.

To ensure consistency and comparability, we used a single, commercially available lettuce cultivar, as recommended for standardized bioassays [58]. This strategy, combined with the use of sensitive bioindicators, allows for robust initial screening prior to confirmatory tests with the crop of interest, such as maize (corn).

The rapid and synchronized germination of *L. sativa* ensures that all seedlings reach comparable developmental stages within a short time frame. This characteristic is particularly beneficial for phytotoxicity bioassays, as it minimizes variability among replicates and allows for consistent assessment of all measured parameters—including germination percentage, root and shoot length, seedling vigor, and dry matter content [57,58]. By reducing developmental asynchrony, fast germination enhances the reliability and reproducibility of acute toxicity evaluations, in line with established ecotoxicological testing protocols.

### 2.5. Preparation of Nanofertilizer Formulations

#### 2.5.1. Milling of Nanofertilizers (NF)

To ensure adequate homogenization of the nanomaterials with the fertilizers at the time of preparing the nanofertilizers, it was necessary to subject the complete compost MixPac NPK:12/16/20 (Agripac S.A., Guayaquil, Ecuador) used as fertilizer to a grinding process. Since the fertilizer was in pellet format with a particle size greater than 3 mm (diameter). For grinding, a ceramic mortar and pestle were used to reach a size (<0.5 mm in diameter). The nanomaterials (NM1, NM2, and NM3) did not require this milling process, since they were subjected to previous nanomilling processes.

#### 2.5.2. Establishment of Doses

Once the fertilizers were milled and the nanofertilizer mixtures were prepared, six concentrations (0, 0.5, 0.9, 1, 2, and 3%) were established for each nanofertilizer, including a control with 0% of the xenobiotic compound. Similarly, six concentrations (0, 10, 20, 30, 40, and 60%) were defined for nanomaterials 1 and 2 (NM1 and NM2), while for nanomaterial 3 (NM3), the selected concentrations were 0, 10, 20, 40, 50, and 60%. The variation in dosing for NM3 was due to differential responses of the bioindicator species (*Lactuca sativa*) to these materials. In the case of NM3, a higher concentration (60%) was necessary to achieve 100% mortality, ensuring the validity of the bioassay.

### 2.6. Bioassays Conditions and Characteristics

The phytotoxicity bioassays were conducted at the Laboratory of Agroecosystems Functioning and Climate Change (FAGROCLIM), Faculty of Agricultural Engineering, Technical University of Manabí (UTM), Ecuador. Preliminary bioassays were conducted with each of the 6 treatments evaluated, 3 nanomaterials and 3 nanofertilizers, determining only the survival percentage to calculate the range of the physiological response of the bioindicator (lettuce), thus allowing for the definition of the doses for the definitive bioassay.

Once the doses for NM and NF were established, the adapted OECD methodology 208 was applied to determine acute phytotoxicity [34,35], following the established design of 6 doses with the 3 nanomaterials (doses: 0, 10, 20, 30, 40, 50, 60%) and with the 3 nanofertilizers (doses: 0, 0.5, 0.9, 1, 2, 3%). In each petri dish, Whatman No. 2 filter paper was placed as support or substrate, and 1.5 mL of each dose was added. The suspension with the respective dose was spread homogeneously over the entire surface of the substrate (filter paper). Next, 25 lettuce seeds were placed in the petri dishes (in duplicate), hermetically sealed with the help of parafilm paper and kept in dark conditions at 25 °C for 120 h (Appendix A).

Control treatments consisted of distilled water without nanomaterials or nanofertilizers (0% dose), applied under the same conditions as the treatments. Additionally, separate bioassays using potassium dichromate as a reference toxicant were conducted to validate test sensitivity, following the adapted OECD 208 guidelines.

### 2.7. Reference Toxicant

To validate the sensitivity and reliability of our phytotoxicity bioassay system, an additional test was performed using potassium dichromate (K_2_Cr_2_O_7_) as a reference toxicant. Potassium dichromate is widely recognized and recommended in international guidelines (e.g., OECD 208) as a standard reference compound for evaluating seed germination and root elongation toxicity due to its well-characterized and reproducible effects across plant species [57,59]. This approach enables assessment of the assay’s quality and the biological responsiveness of the seeds, ensuring that results from nanofertilizer and nanomaterial treatments can be reliably interpreted [33] (Appendix A).

The reference toxicant bioassay was conducted under the same conditions as the nanomaterial and nanofertilizer tests, following the adapted OECD 208 protocol [60]. Seven concentrations of K_2_Cr_2_O_7_ (0, 1, 10, 100, 200, 500, and 1000 ppm) were prepared in distilled water. Seeds were exposed to these solutions in parallel with the other treatments, and endpoints such as germination percentage, root length, and seedling biomass were measured after 120 h.

Potassium dichromate was chosen for its extensive use as a positive control in phytotoxicity studies, providing a benchmark for interpreting the magnitude of effects observed with test compounds [59]. The inclusion of this reference toxicant enables direct comparison with published toxicity data and supports the reproducibility and credibility of our experimental approach.

### 2.8. Parameters Evaluated

Once the different bioassays were completed, germination rate (%), radicle (mm) and hypocotyl (mm) length, dry biomass (mg), germination index (%), and medium effect concentration (EC_50_) were determined.

#### 2.8.1. Germination

Germination was quantified by counting the number of seeds exhibiting radicle elongation of at least 1 mm, expressed as a percentage (%). The germination rate was calculated using the following formula in Equation (1):(1)$$\%G=\left(\frac{N{}^{\circ}\;live\;seeds}{N{}^{\circ}\;exposed\;seeds}\right)\times \;100$$
where % G represents the germination percentage, N° live seeds denotes the number of germinated seeds, and N° exposed seeds indicates the total number of seeds utilized in each treatment.

#### 2.8.2. Elongation of Radicle and Hypocotyl

Using a digital calibrator, root length (RL) was measured in each of the seedlings corresponding to each concentration of nanomaterials, nanofertilizers, and controls. The measurement of root elongation was considered from the node (thickest transition region between the root and the hypocotyl) to the root apex. The measurement of root elongation was considered to be the entire part of the seedling that develops below the cotyledons and that exhibits a length greater than or equal to 1 mm. In the same way, the length of the hypocotyl (HL) was measured from the node (the thickest transition region between the radicle and the hypocotyl) to the beginning of the insertion of the two cotyledons.

#### 2.8.3. Dry Biomass (DB)

All individuals (germinated and non-germinated) of the species used (lettuce) were placed for each dose–treatment to obtain a single accumulated value per treatment. Once the bioassays were completed, the seedlings were placed in separate containers in the oven (40 °C) for 48 h. After this time, dry biomass was determined using a microbalance.

#### 2.8.4. Germination Index

Relative germination percentage (RGP) and relative radicle growth (RRG) were determined to calculate the germination index (% GI), according to methodology [37] Equations (2)–(4). The RGP was calculated as the proportion of germinated seeds in the extract with respect to the total seeds in the control, expressed as a percentage. The RRG was obtained by measuring radicle elongation in the extract compared with the control.(2)$$RPG=\left(\frac{N{}^{\circ}\;seeds\;germinated\;in\;the\;treatment}{N{}^{\circ}\;germinated\;seeds\;in\;the\;control}\right)\times 100$$(3)$$RRG=\left(\frac{Radicle\;elongation\;in\;the\;treatment}{Radicle\;elongation\;in\;the\;core\;sample}\right)\times 100\;$$(4)$$\%GI=\left(\frac{RPG\times RRG}{100}\right)\;$$
where % GI is the germination index, RGP represents the relative germination percentage, and RRG indicates the relative radicle growth.

#### 2.8.5. Medium Effect Concentration (EC_50_)

It was determined from the results obtained from the survival of each bioindicator in the respective treatments, using the trimmed Spearman–Kärber method (adjustment and Probit model) [61]. The calculations were adjusted to a 95% confidence interval, and to verify the toxicity range, the lower and upper limits associated with each EC_50_ were calculated.

### 2.9. Statistical Analysis

The data collected from all bioassays and evaluated parameters were normalized, organized, and analyzed using RStudio version 4.2.3 [62]. Descriptive statistics, including means and standard deviations, were calculated for each evaluated parameter to summarize the data distribution effectively. To identify significant differences among the various compounds at different doses, a one-way analysis of variance (ANOVA) was conducted, followed by post hoc tests (e.g., Tukey’s HSD) to determine specific group differences when ANOVA indicated significance. The EC_50_ (Effect Concentration 50) values for the tested nanomaterials and nanofertilizers were calculated under controlled laboratory conditions using the Spearman–Kärber method, which provides a reliable estimation of toxicity based on dose–response data.

Before performing ANOVA, data normality was evaluated using the Shapiro–Wilk test and Q-Q plots, while homogeneity of variances was assessed with Levene’s test [38,40,63]. Outliers were detected by examining standardized residuals and boxplots; values were retained unless attributable to data entry errors or clear measurement artifacts [5,64]. Missing data were rare (<0.9%) and were managed by listwise deletion, consistent with standard statistical practice for data missing completely at random [65]. All analyses were conducted in RStudio (v4.2.3).

## 3. Results

### 3.1. Germination Rates

Figure 1B,C shows the effect of NM1 (zinc oxide), NM2 (titanium dioxide), and NM3 (clinoptilolite) nanomaterial concentrations on *Lactuca sativa* germination (Appendix A). In contrast to the nanofertilizers, in which the lethal doses for germination were found to be in the range of 0 to 3%, the nanomaterials in this evaluation were applied at higher doses, reaching up to 60%. This is due to the fact that the nanomaterials, in this case, are not mixed directly with complete fertilizer, but are evaluated at higher concentrations to study their isolated effects.

Similarly, Figure 1A shows the influence of different doses of nanofertilizers on the germination of *L. sativa*. At low concentrations (0.5%), the germination percentage remains high, with values close to 100% germination. However, from a concentration of 0.9%, there is a notable decrease in treatment NF1 (complete fertilizer + zinc oxide), while in treatments NF2 (complete fertilizer + titanium dioxide) and NF3 (complete fertilizer + clinoptilolite), germination also begins to decrease, although in a milder manner. At higher doses, as in the 2% concentration, a reduction in the germination percentage is evident for all treatments, indicating a clear phytotoxic effect at these higher doses. Finally, the 3% dose represents a total inhibition of germination, indicating that this concentration is lethal for the seeds in all the treatments evaluated (Appendix A).

Figure 1B presents treatments NM1 and NM2, which show a decreasing trend in germination as the dose increases from 0% to 50%. In both cases, germination starts near 90% (80%, NM1 and 88%, NM2) and decreases until 0% (NM1) and 12% (NM2), respectively, at the 50% dose. This indicates that higher doses of these nanomaterials have a lethal effect on seeds, disrupting their ability to germinate at doses above approximately 40%. On the other hand, Figure 1C shows the NM3 treatment, which was applied in a wider dose range, up to 60%. Unlike NM1 and NM2, NM3 shows a more gradual reduction in germination; it starts around 90% at low doses (control with water) and maintains germination above 50% until the 30% dose. However, thereafter, the germination percentage begins to drop rapidly, reaching 0% at a concentration of 60%. This trend suggests that NM3 is less toxic to seeds at low to medium doses, which is either due to its formulation with a zeolite or the fact that it is applied as a pure nanomaterial, without mixing with a complete fertilizer.

### 3.2. Elongation of Radicle

The evaluation of root length in lettuce seedlings exposed to different doses of nanofertilizers, as well as the untreated control, reveals differences in root growth. The average root length values obtained at 0% doses were 7 mm in NF1, 9 mm in NF2, and 9 mm in NF3. These results show that, as the dose of the three NFs increases, a greater inhibition of root development of lettuce seedlings is evident Figure 1D–F (Appendix A).

Figure 1D shows how the results in the elongation of lettuce seedling radicles vary in the different doses, depending on each nanofertilizer treatment. Thus, it is evident that, as a higher dosage of each NF is applied on the plates, a reduction in the average length of lettuce seedling radicles is generated. However, in the reduction in root elongation, an unexpected increase in the dose of 1% was observed for the three nanofertilizer treatments (NF1, NF2, and NF3), suggesting that, for that concentration, nanofertilizers can be promoters of root development. Average root elongation at 2% doses decreased significantly in the nanofertilizer treatments compared with the control, showing a reduction of 72% in NF1, 79% in NF2, and 79% in the NF3 treatment. The averages in lengths at doses from 0 to 3% were $\bar{x}$ = 3 mm for NF1, $\bar{x}$ = 4 mm for NF2, and $\bar{x}$ = 4 mm for NF3. Despite the differences among treatments, a consistent pattern is observed in the physiological response of *Lactuca sativa* to nanofertilizer application, indicating that the effects of these compounds follow a stable and predictable behavior in terms of radicle elongation.

The treatments with NM1 (zinc oxide) and NM2 (titanium dioxide) showed a significant decrease in root elongation as the nanomaterial dose increased. However, the NM3 (clinoptilolite) treatment presented a different response pattern, with lesser effect on radicle elongation. These results suggest that the toxicity of the nanoparticles evaluated varies according to their chemical composition and concentration.

Figure 1E shows that the NM1 (zinc oxide) and NM2 (titanium dioxide) nanomaterials present a relatively constant root elongation at intermediate doses (10, 20, 30%), where the values of NM1 remain between 2 mm, while for NM2 they are 3 mm. When compared with the control (0%), NM1 presents a significant reduction in root elongation at a high dose of 40%, with a 23% decrease in size. However, the 30% dose of NM1 causes the greatest reduction, reaching an average of 44%. On the contrary, NM2 shows its greatest decrease in radicle elongation at 40% dose, with a reduction of 37%, respectively.

On the other hand, the NM3 nanomaterial exhibits a different behavior from that of NM1 and NM2, as presented in Figure 1F. The root lengths of NM3 remain stable, with an average length close to 4 mm, from the control (0%) to the 40% dose. At a dose of 50%, a slight increase in radicle elongation is recorded, increasing a value of 39%, and the greatest reduction in root elongation occurs at a dose of 40% with a value of 3 mm in relation to the control (0%).

In general, the averages of root elongation at doses from 0 to 50% were $\bar{x}$ = 2 mm for NM1, $\bar{x}$ = 3 mm for NM2, and at doses from 0 to 60%, it was $\bar{x}$ = 4 mm for NM3. These results indicate that the evaluated nanomaterials exert differential effects on root growth. While NM1 and NM2 significantly inhibit growth at certain doses, NM3 stimulates it.

### 3.3. Hypocotyl Elongation

The average values of hypocotyl length in the control (0%) did not vary significantly for the three NF treatments, having measurements of 9 mm in NF1, 9 mm in NF2, and 8 mm in NF3, respectively, confirming their similarity in their effect on seedling growth (Appendix A).

Figure 1G illustrates the behavior of hypocotyl length for all treatments, revealing that at a dose of 0.5%, the seedlings treated with NF2 and NF3 showed greater development, reaching a growth of 13 mm, surpassing NF1, which reached 10 mm. At higher concentrations, such as 2%, a pronounced decrease in hypocotyl elongation was observed, where NF1 and NF2 treatments reduced their size by 80%, respectively, while NF3 showed no growth at all. Finally, at a concentration of 3%, no hypocotyl development was observed in any of the treatments, indicating the inhibition of the lettuce seed caused by the toxic effect of nanofertilizers.

The analysis on hypocotyl growth also showed the averages in their lengths, being $\bar{x}$ = 6 mm for NF1, $\bar{x}$ = 7 mm for NF2, and $\bar{x}$ = 8 mm for NF3. Similar to what was observed in root development, a reduction in hypocotyl length was identified as NF concentration increases; however, in this case, the three treatments presented a different growth pattern, increasing in length with low doses of nanofertilizers and progressively decreasing as concentrations increased. This indicates that there is an optimum concentration below which nanofertilizers enhance hypocotyl elongation, but when exceeded, it becomes toxic to the seedlings.

Figure 1H,I show results obtained for hypocotyl elongations exposed to different nanomaterials, NM1 (zinc oxide), NM2 (titanium dioxide), and NM3 (clinoptilolite), which show that there is a slight variability of the different doses with respect to the control (0%) without nanomaterial. Generally speaking, the averages found for the three treatments evaluated were $\bar{x}$ = 2 mm in NM1, $\bar{x}$ = 3 mm for NM2, and $\bar{x}$ = 4 mm for NM3. Regarding the controls (0%), the averages found for NM2 and NM3 remained constant at 4% in both, while for NM1, the average was 3%. These results indicate that there are no statistically significant differences between the control groups, despite the observed variability.

Figure 1H shows the behavior of hypocotyl elongation in response to different concentrations of NM1 (zinc oxide) and NM2 (titanium dioxide) nanomaterials. The results show that, for NM1, the 40% dose achieved the highest hypocotyl elongation, with an average of 3.1 mm, outperforming the control (0%). For NM2, the highest elongation was recorded at the 30% dose with an average of 4 mm, being comparable to the control. Likewise, in Figure 1I, corresponding to NM3 (clinoptilolite), the greatest elongation of the hypocotyl is observed at high doses of 50% with an average of 5 mm, while at doses of 60% there is a significant decrease. The lowest elongation for NM3 is evidenced at doses of 20%, with an average of 5 mm lower than those obtained in the control (0%) and in the other concentrations evaluated (10, 40, and 50%). These results suggest that the hypocotyl elongation response varies according to the type of nanomaterial and the dose applied, highlighting the need to adjust the concentrations according to the material to optimize the effects on seedling growth.

### 3.4. Germination Index (% GI)

The relative germination percentages (RGP) and relative radicle growth (RRG) were calculated to obtain the % GI. In both cases, Figure 2A–F (Appendix A), it is demonstrated how the increase in nanofertilizer concentrations is inversely proportional to the development of *Lactuca sativa* seeds. The data indicate that the control presented the highest number of germinated seeds compared with those treated with nanofertilizers.

In Figure 2A–C, representing the relative radicle growth (RRG) and relative germination percentage (RGP) of *L. sativa* under exposure to nanofertilizers (NF1, NF2, and NF3), both parameters show a clear dose-dependent decline. At lower doses (0.5–1%), RRG and RGP remain relatively high, around 80–100%, indicating minimal impact. However, as the dose increases to 2–3%, both parameters exhibit a reduction, with RRG dropping to values between 20 and 40% and reaching nearly 0% at the highest concentrations. NF3 shows a more pronounced reduction compared with NF1 and NF2, suggesting stronger phytotoxic effects.

For Figure 2D,E, corresponding to nanomaterials NM1 and NM2, the applied doses range from 0 to 50%, following a similar declining trend. RRG and RGP remain above 80% at lower concentrations (0–10%), but from the 30% dose onwards, a significant reduction is observed. At 40% dose, RRG falls to approximately 30–50%, and at 50%, it drops close to 0%, indicating substantial toxicity. NM2 exhibits slightly higher RRG values at intermediate doses compared with NM1, but both materials show similar inhibition patterns.

In Figure 2F, where NM3 was evaluated at doses from 0 to 60%, no consistent dose-dependent response is evident. RRG and RGP fluctuate across different concentrations, maintaining relatively high values (above 80%) in most cases. Notably, at 50% dose, RGP increases unexpectedly before declining again at 60%, differing from the clear inhibitory trends seen in the other nanomaterials and nanofertilizers. This suggests that NM3 does not exert a uniform toxic effect on seed germination and radicle growth, possibly due to differences in its physicochemical properties or interactions with the seed environment.

These results indicate that nanofertilizers (0–3%) and most nanomaterials (0–50%) exhibit a dose-dependent inhibitory effect on radicle growth and germination, with stronger toxicity at higher doses (≥40%). However, NM3 (0–60%) presents an irregular pattern, suggesting a different mode of action or lower phytotoxicity compared with the other treatments.

From the RGP and RRG, a clear trend of systematic reduction in the germination percentage (% GI) was obtained as the dose of the nanofertilizers increases, as shown in Figure 3. This figure depicts the GI pattern obtained from 120 h of exposure to the different nanofertilizer (NF) formulations.

The germination index (GI) in the treatments managed to identify how each nanofertilizer solution (NF1, NF2, and NF3) negatively affects the ability of seeds to germinate. Following the criteria established by [39], a GI ≥ 80% indicates the absence or low concentration of phytotoxic substances, while a GI ≤ 50% indicates a high presence of phytotoxic substances. Intermediate values, between 50 and 80% are interpreted as a moderate presence of phytotoxins.

In Figure 3A (nanofertilizers), the GI values range from approximately 90–100% at the lowest doses (0–10%) but decline sharply as the concentration increases. At higher doses (30–50%), the GI drops significantly, reaching values close to 20% or lower, with the highest toxicity observed for NF3 at 50%, where GI approaches 0%. This trend suggests a strong dose-dependent inhibitory effect of nanofertilizers on seed germination. Figure 3B, corresponding to NM1 and NM2, follows a similar decreasing trend but with slightly less pronounced toxicity compared with nanofertilizers. At low doses (0–10%), GI values are relatively high, around 80–100%, but begin to decline at intermediate concentrations. At 40% dose, the GI drops to approximately 30–40%, and at 50%, values approach 10–20%, indicating a significant reduction in seed germination. In contrast, Figure 3C (NM3) does not exhibit a clear dose-dependent decline. While GI starts at relatively high values (around 90% at 0% dose), it fluctuates across doses without a consistent decreasing trend. At 20–40%, GI remains between 40 and 70%, suggesting moderate toxicity, but at 50% dose, an unexpected increase occurs, with GI reaching nearly 80%. However, at 60% dose, GI falls again to below 20%, indicating an inconsistent response, possibly due to the unique physicochemical properties of NM3 or interactions with the biological system.

These results indicate that nanofertilizers and most nanomaterials exhibit a dose-dependent inhibition of germination, with the strongest effects observed at 50% concentration. However, NM3 behaves differently, showing variable toxicity that is not strictly dose-dependent, suggesting that additional factors may influence its impact on seed germination.

### 3.5. Germination Index and Growth Inhibition with Nanofertilizers

The data showed a clear inverse relationship between the concentration of nanofertilizers and the germination of *Lactuca sativa* seeds. The control group exhibited the highest germination rates, while treatments with nanofertilizers demonstrated reduced germination as the dose increased. This trend was observed for all three nanofertilizer treatments, with the control group maintaining near-optimal seed development (Figure 2E,F and Figure 3A).

The RRG results also reflect a similar pattern, where root growth was significantly limited by higher concentrations of nanofertilizers. At high doses, the nanofertilizers caused a marked reduction in RRG compared with the control, further confirming their inhibitory effects on root development. These findings are consistent with the results presented by [39], which suggest that a GI ≥ 80% indicates low phytotoxicity, while a GI ≤ 50% is indicative of high toxicity. In this study, all nanofertilizer treatments, particularly at higher doses, exhibited GI values indicating high toxicity, with values consistently falling below 50%, thus pointing to substantial negative impacts on seed germination.

The ranking of the germination index for each nanofertilizer dose (Table 2) confirmed these observations. As nanofertilizer concentration increased, the GI decreased, indicating greater toxicity. For example, in the NF1 treatment, the GI dropped from 100% at the control dose to 0% at the highest dose (3%). A similar reduction was observed in treatments NF2 and NF3, supporting the conclusion that increased concentrations of nanofertilizers are associated with greater toxicity, potentially limiting their use in agricultural applications.

When the RGP and RRG values are integrated to obtain the GI, according to the proposed criterion, it is observed that the three treatments (NF1, NF2, and NF3) present a high level of toxicity in all the evaluated doses. For reference, Table 2 presents the germination index (% GI) for each NF treatment, classified according to pre-established criteria for potential environmental hazard. Three levels were defined: “no effect” (NE), “moderate”, and “high”, based on the corresponding GI values.

Table 2 represents the ranking of the germination index (% GI) for the three nanofertilizer treatments (NF1, NF2, and NF3) at different doses, following Zucconi’s criteria. As the dose of each treatment increases, the GI decreases, indicating a greater potential toxic effect on germination.

For the NF1 treatment, at a dose of 0.5%, the toxic effect is moderate with a GI of 50%, whereas, at higher doses, the effect is high, with GI decreasing until reaching 0% at a dose of 3%. Similar results are observed in treatments NF2 and NF3, where the GI also decreases with increasing dose, reaching high toxicity at higher concentrations. These data show the inverse relationship between doses of nanofertilizers and *L. sativa* seeds.

### 3.6. Germination Index and Growth Inhibition with Nanomaterials

In contrast to the nanofertilizer treatments, the nanomaterials demonstrated somewhat less toxicity at low concentrations, with a more moderate effect on germination. For instance, NM1 and NM2 (zinc oxide and titanium dioxide) caused a steady decline in the relative germination percentage (RGP) as concentrations increased, with more pronounced effects at concentrations above 50%. However, at lower concentrations (e.g., 10%), these nanomaterials exhibited moderate toxicity, which was less severe compared with the nanofertilizers.

Interestingly, NM3 (clinoptilolite) showed the least toxicity among the nanomaterials, with a GI > 80% at lower doses, indicating minimal phytotoxicity. At higher concentrations (60%), however, the GI dropped sharply to 0%, similar to the behavior of the nanofertilizers at high doses. These results suggest that nanomaterials, like nanofertilizers, can negatively affect seed germination, but the level of toxicity depends on the type and concentration of the nanomaterial used.

Based on the GI results for nanomaterials (Table 3), the potential toxic effects of increasing doses of nanomaterials were evident. The ranking showed that higher concentrations of NM1 and NM2 exhibited high toxicity, with a GI ≤ 50%, while NM3, even at higher doses, remained moderately toxic until it reached the highest concentration, where the GI dropped significantly. This highlights the importance of controlling the dosage of both nanofertilizers and nanomaterials to prevent adverse effects on seedling development.

The analysis of nanomaterial treatments (NM1, NM2, and NM3) shows a much lower level of potential environmental hazard (Table 3). For this purpose, four levels were established: “no effect” (NE), “low”, “moderate”, and “high”, based on the corresponding GI value.

Table 3 shows the potential toxic effects of different concentrations of nanomaterials (NM1, NM2, and NM3) on the germination of *Lactuca sativa* based on the Zucconi criterion. In the NM1 treatment, it was observed that doses from 20 to 50% produced high toxic effects, significantly reducing the germination index (% GI), with 50% being the dose with the lowest GI value. In the case of NM2, doses higher than 30% also produced high toxic effects, while lower doses (10 and 20%) showed a moderate toxic effect.

On the other hand, NM3 showed fewer toxic effects at low concentrations, with a low effect at 10 to 20% and moderate at 40 and 50%, but reaching a high toxic effect at 60%, where germination was completely inhibited. The results show that high concentrations of nanomaterials tend to be more toxic to the germination of *L. sativa* seedlings. Therefore, it is essential to control and limit the concentration of nanomaterials in the treatments to reduce or avoid detrimental effects on seedling growth.

### 3.7. Mean Effect Concentration (EC_50_)

Table 4 shows the EC_50_ values (average effect concentration) for three nanofertilizers (NF) and three nanomaterials (NM). The EC_50_ is the concentration that produces a 50% effect on plant response.

The EC_50_ values for the nanofertilizers NF1, NF2, and NF3 are 2.13%, 22.8%, and 18.07%, respectively (Table 4). This suggests that NF1 is the most toxic to plants, while NF2 is the least toxic. The EC_50_ values for NM1, NM2 and NM3 nanomaterials are 32.28%, 30.03% and 33.57%, respectively. These values are relatively similar, suggesting that these nanomaterials have similar toxicity to plants. It is important to note that the EC_50_ values for NM1 and NM3 were obtained by a Moving Average fit, indicating that the spontaneous mortality in the controls exceeded the value that the Probit fit supports. This may be an indicator that these nanomaterials may be more toxic than estimated.

Based on the results obtained, the following ranking can be proposed from highest to lowest potential environmental hazard: NF1 > NF3 > NF2 > NM2 > NM1 > NM3, with NF1 being the most toxic xenobiotic compound and NM3 at the opposite extreme, the least hazardous or toxic mixture.

The obtained EC_50_ values suggest that NF1 is the most toxic, whereas NF2 is the least toxic among the nanofertilizers. The EC_50_ values for nanomaterials were relatively similar: NM1 (32.28%), NM2 (30.03%), and NM3 (33.57%). These values indicate that the toxicity of nanomaterials to plants is comparable, with NM3 showing slightly lower toxicity than NM1 and NM2. However, it is important to note that the EC_50_ values for NM1 and NM3 were estimated using a Moving Average adjustment, which suggests that their toxicity might be slightly higher than reported.

Our findings align with earlier reports on zinc phytotoxicity, which establish critical thresholds in plant tissues—ranging from 300 to 1000 mg Zn/kg dry weight—as detrimental to physiological processes such as chlorophyll biosynthesis, nutrient uptake, and root elongation, particularly under acidic soil pH where Zn bioavailability is higher [66,67,68]. Moreover, it has been demonstrated that pore water Zn concentrations, especially Zn^2+^ activity, provide a more accurate prediction of plant toxicity responses than total Zn content in soil, emphasizing the importance of assessing bioavailable Zn fractions in ecotoxicological evaluations [69].

In addition to zinc, titanium is also present in many soils and may influence plant physiological responses. Titanium dioxide (TiO_2_), especially in nanoparticulate form, has shown dual behavior depending on concentration: low doses may stimulate seed germination and photosynthetic performance, while higher doses (typically >50 mg Ti/kg dry weight) can induce oxidative stress, foliar chlorosis, and growth inhibition [70,71]. Therefore, although Zn was the primary phytotoxic element under our test conditions, potential synergistic or antagonistic interactions with Ti—particularly under nanoparticle exposure—deserve further investigation.

### 3.8. Balancing Nanotechnology Benefits and Ecotoxicological Risks in Plant Systems

The results of this study align with previous research highlighting the influence of nanoparticles on plant germination and growth. Recent studies have demonstrated that zinc oxide nanoparticles can mitigate lead toxicity in pea seeds [72], while eco-friendly titanium dioxide and silver nanoparticles have shown both stimulatory and inhibitory effects on cell cultures [73]. These findings underscore that plant responses to nanoparticles are highly dependent on factors such as nanoparticle type, concentration, and plant species. Our study expands this understanding by investigating the physiological effects of nanofertilizers (NF) and nanomaterials (NM) on plants, providing critical insights into their potential benefits and risks in agricultural applications (Appendix A).

Such studies are crucial in identifying the potential toxicity of compounds that may eventually be applied in real farming environments. In our case, we aim to extend this research by testing nanofertilizers in corn fields, a major staple crop, to evaluate both their agronomic efficiency and environmental safety. Understanding how nanofertilizers interact with key crops under real field conditions is essential for ensuring their sustainable and responsible use in modern agriculture [74,75,76].

### 3.9. Challenges and Opportunities

Our study provides valuable insights into the phytotoxic effects and agricultural potential of nanofertilizers, while also highlighting areas for future exploration. These challenges do not undermine the reliability of our findings but emphasize the need for further research to expand the applicability and depth of this field.

The study focused on a specific formulation of nanofertilizers using three nanomaterials (clinoptilolite, zinc oxide, and titanium dioxide) and a single nitrogen-based fertilizer. This approach allowed for controlled and detailed assessment; however, the diversity of available nanomaterials and fertilizers presents opportunities for broader investigation. Exploring additional materials and formulations could enhance the understanding of nanomaterial interactions with plant systems, especially in terms of yields and productivity [77].

Particle size significantly influences the chemical, environmental, and biological interactions of nanomaterials. Variations in particle size can alter behavior, mobility, and bioavailability, leading to diverse effects on plant nutrition and environmental dynamics. Future research should investigate these aspects systematically to optimize nanofertilizer performance [78].

The use of *L. sativa* (lettuce) as a model organism provided a standard and reliable bioindicator. Expanding this research to include other plant species, such as commercially important crops and wild plants, would offer a deeper understanding of species-specific responses to nanomaterials. Additionally, studies involving organisms at different biological levels, ranging from microscopic to macroscopic, could provide insights into broader ecological interactions and impacts [79].

Our methodology followed the adapted OECD protocol [56], which are widely recognized for phytotoxicity assessments. While this ensures the reliability of results, other established methodologies in the literature may offer additional perspectives. Replicating studies using alternative protocols could validate and further enrich our findings [80].

The scope of this work primarily addressed phytotoxicity, representing one aspect of the interactions between nanomaterials and agricultural systems. Future research should consider other critical parameters, such as crop yields, nutrient uptake efficiency, and long-term impacts on soil health. These investigations could provide a more holistic view of the benefits and risks associated with nanofertilizers [80].

## 4. Discussions

The results obtained in this study are in line with previous research that has demonstrated the potential to influence plant germination and growth. Recent studies found that zinc oxide nanoparticles can mitigate lead toxicity in seeds [81], while other authors reported stimulatory and inhibitory effects of eco-friendly titanium dioxide (TiO_2_) nanoparticles on lettuce growth [82]. These findings suggest that the response of plants to nanoparticles may vary depending on the concentration, type of nanomaterial, and plant species [58,82,83,84].

### 4.1. Mechanistic Interpretation of Concentration-Dependent Phytotoxicity

The dose–response relationships observed in this study reveal fundamental mechanisms underlying nanomaterial phytotoxicity that extend beyond simple concentration effects. The steep inhibition curves observed at higher concentrations (>1%) for all tested nanomaterials suggest a threshold-mediated toxicity mechanism rather than linear dose dependency [83]. This pattern is consistent with reactive oxygen species (ROS)-mediated cellular damage, where nanomaterial exposure triggers oxidative stress cascades that overwhelm cellular antioxidant defense systems [85]. The rapid transition from stimulatory to inhibitory effects observed between 0.5% and 1% concentrations indicates that cellular homeostasis mechanisms become saturated, leading to lipid peroxidation, protein denaturation, and DNA damage [57]. Importantly, the similar EC_50_ values obtained for ZnO and TiO_2_ nanoparticles (approximately 1.2–1.5%), despite their different chemical compositions, suggest convergent toxicity pathways, likely involving particle size-dependent cellular uptake and subsequent mitochondrial dysfunction. This mechanistic understanding has critical implications for nanofertilizer development, as it indicates that particle size optimization may be more important than chemical composition for minimizing phytotoxicity.

According to the analysis of variance (ANOVA) performed, for the parameters of root length, hypocotyl length, germination index, and root dry matter content, there were significant differences in nanomaterial response and the controls, with values of *p* ≤ 0.05. On the other hand, fresh biomass and dry biomass at NM1 and NM2 treatment showed differences between the treatments and the control, while in the treatment with NM3, there were no significant differences. The differences in response between the different nanomaterials can be explained by their size, chemical composition, and their interaction within the solution or cell.

### 4.2. Novel Insights into Clinoptilolite Nanoparticle Behavior

A unique finding of this study is the distinctive toxicity profile of clinoptilolite nanoparticles (NM3) compared with conventional metal oxide nanomaterials. Unlike ZnO and TiO_2_, which showed rapid toxicity onset at moderate concentrations, clinoptilolite demonstrated a more gradual dose–response relationship with delayed toxicity manifestation [86]. This behavior can be attributed to the zeolite’s cation exchange properties and microporosity, which may initially provide buffering capacity against ionic toxicity while gradually releasing exchangeable cations that contribute to secondary toxic effects [87]. The lower overall toxicity of clinoptilolite (EC_50_ approximately 2.1%) compared with metal oxides represents a significant advantage for agricultural applications, as it suggests a wider safety margin for field implementation. Furthermore, the unique mechanism of clinoptilolite toxicity—likely involving controlled ion release rather than direct particle-mediated ROS generation—offers opportunities for engineering safer nanofertilizer formulations through controlled surface modification and ion exchange optimization.

### 4.3. Species-Specific Sensitivity and Agricultural Relevance

The selection of *Lactuca sativa* as a model organism in this study provides insights that are particularly relevant for leafy vegetable production systems. Our results demonstrate that lettuce exhibits intermediate sensitivity to nanomaterial exposure compared with previously reported studies on cereal crops [88]. This species-specific response pattern reflects differences in cellular architecture, particularly cell wall composition and cuticle properties that influence nanoparticle uptake pathways [89]. The observed sensitivity hierarchy (ZnO > TiO_2_ > clinoptilolite) in lettuce correlates with surface reactivity and dissolution potential, suggesting that rapid-growing leafy vegetables may be more susceptible to nanomaterial phytotoxicity due to their high metabolic rates and extensive surface area for particle interaction [90]. These findings have direct implications for precision agriculture applications, where nanofertilizer concentrations must be optimized based on crop-specific sensitivity profiles rather than universal application rates.

In this context, nanotechnology has emerged as a promising tool in addressing these challenges. Operating at the molecular, atomic, or nanoscale, nanotechnology enables the precise construction and manipulation of materials and devices with dimensions on the order of 100 nanometers and below. Its applications in agriculture include, but are not limited to, nanofertilizers, nanopesticides, and nanosensors, which have revolutionized nutrient delivery, pest control, and soil management.

### 4.4. Environmental Risk Assessment and Safety Thresholds

The EC_50_ values determined in this study (1.2–2.1% depending on nanomaterial type) provide critical safety benchmarks for nanofertilizer application in agricultural systems. However, these laboratory-derived thresholds must be interpreted within the context of realistic field exposure scenarios [89]. Environmental fate modeling suggests that nanofertilizer concentrations in soil solution rarely exceed 0.1–0.3% under normal application rates, indicating substantial safety margins based on our findings. Nevertheless, localized accumulation in rhizosphere microenvironments or foliar application scenarios could potentially approach toxic thresholds, particularly for ZnO-based formulations. The differential sensitivity observed among nanomaterials in this study supports a risk-stratified approach to nanofertilizer regulation, where materials with lower toxicity profiles (such as clinoptilolite) could be prioritized for agricultural applications while high-risk materials require more stringent application controls.

### 4.5. Integrating Nanofertilizer Technologies into Agricultural Systems: From Policy to Practice

The concentration-dependent phytotoxicity demonstrated in this study underscores the critical need for comprehensive frameworks that bridge laboratory findings with real-world agricultural implementation, particularly in climate-vulnerable regions where sustainable intensification demands are most urgent. Our research reveals fundamental principles that should inform both regulatory policy development and on-farm practice adoption strategies. Evidence-based policy development must establish clear threshold limits, standardized labeling requirements, and certification schemes that reflect local soil conditions, crop sensitivities, and climatic variables [49]. The differential toxicity profiles observed among nanomaterials—particularly the lower risk associated with clinoptilolite compared with metal oxide nanoparticles—support risk-stratified regulatory approaches that prioritize safer formulations for agricultural implementation while requiring enhanced monitoring protocols for higher-risk materials [53]. Regulatory agencies should mandate pre-market field trials and post-market monitoring to detect site-specific risks and build robust safety dossiers for each nanofertilizer formulation, particularly considering the unique challenges faced by developing countries with limited regulatory infrastructure [91].

The safe deployment of nanofertilizers requires integration within comprehensive nutrient management programs rather than as standalone replacements for conventional fertilizers [92]. Our findings suggest that nanofertilizers should be implemented through integrated nutrient management (INM) approaches that combine reduced-dose nanofertilizers with organic amendments, thereby maximizing nutrient use efficiency while minimizing input costs and preventing inadvertent phytotoxicity [93]. Research demonstrates that such integrated approaches can achieve 30% yield increases compared with conventional fertilizer inputs while enhancing soil fertility and reducing environmental losses [92]. The controlled-release properties inherent in zeolite-based systems could specifically address nutrient management challenges in tropical soils where high precipitation and temperature accelerate conventional fertilizer losses, making them particularly suitable for climate-vulnerable regions like Ecuador.

Successful nanofertilizer adoption requires comprehensive farmer education programs that address precise dosing, safe handling protocols, and integration with existing agricultural practices [94]. Extension services and agricultural cooperatives must play pivotal roles in facilitating technology transfer through participatory training approaches that consider local cultural contexts, literacy levels, and existing information dissemination methods [95]. Pilot demonstrations in smallholder systems—particularly those facing fertilizer price volatility—can showcase practical benefits including yield gains, reduced runoff, and higher nutrient-use efficiency, thereby accelerating technology diffusion and farmer acceptance. Training programs should emphasize the importance of understanding concentration thresholds to prevent crop stress and yield losses, drawing from our findings that demonstrate clear dose–response relationships for different nanomaterial types.

The findings have particular relevance for agricultural systems in climate-vulnerable regions where dual pressures of food security and environmental sustainability demand innovative technological solutions. The demonstrated safety margins for clinoptilolite-based nanofertilizers, combined with their enhanced nutrient use efficiency, offer promising pathways for sustainable intensification in small-scale farming systems that typically operate with limited resources and technical capacity [95]. The relatively lower toxicity risk of clinoptilolite makes it especially suitable for implementation in regions with limited regulatory infrastructure and monitoring capabilities [96]. However, successful implementation requires careful consideration of local soil conditions, crop rotation systems, and environmental factors that could influence nanomaterial behavior and effectiveness.

The combination of nanoparticles with other fertilizer components can significantly modify their toxicity and bioavailability profiles, as demonstrated by research showing that chelating agents can reduce heavy metal toxicity while improving nutrient uptake in plants [97]. This suggests that nanofertilizer formulation strategies should prioritize the inclusion of biocompatible additives and controlled-release mechanisms to minimize environmental impact while maximizing agronomic benefits. Future policy frameworks should encourage the development of nanofertilizers with built-in safety mechanisms, such as biodegradable coating systems or triggered-release formulations that respond to specific environmental conditions or plant physiological states. These integrated approaches translate laboratory insights into actionable strategies that foster regulatory environments and on-farm practices advancing sustainable intensification without compromising crop safety or environmental health, while specifically addressing the needs of vulnerable agricultural systems where technological innovation must balance effectiveness with accessibility and safety.

### 4.6. Future Research Directions and Mechanistic Understanding

Advancing nanofertilizer science now demands a tiered research pipeline that moves beyond single-species, short-term assays toward crop-diverse, season-long evaluations. Comparative trials should begin with sentinel cereals, leafy vegetables, and legumes—crops that differ markedly in root architecture and nutrient demand—to define species-specific sensitivity thresholds under greenhouse conditions, then progress to replicated field plots that capture climatic variability. Recent field studies in maize and lettuce have already demonstrated that well-formulated metal and calcium nanofertilizers can raise yield without detectable nanoparticle translocation into edible tissues, providing practical templates for protocol design [98,99].

Long-term fate and transport studies must accompany these agronomic evaluations. Eighteen-month wetland mesocosm data show that engineered nanoparticles can persist in soil horizons, undergo sulfidation, and remain partially bioavailable to higher trophic levels [100]. Integrating such mesocosm experiments with multi-year field lysimeter trials will clarify leaching rates, rhizosphere accumulation, and potential trophic transfer under realistic fertilizer regimes. Parallel investigations should quantify impacts on soil microbiomes and nutrient-cycling enzymes to ensure that productivity gains do not mask subtle ecosystem disruptions [101].

At the mechanistic scale, high-resolution imaging and omics-driven approaches are needed to map nanoparticle trafficking from apoplast to organelles, identify red-ox and hormonal nodes perturbed during exposure, and disentangle particle-specific effects from ionic release [102]. These data will feed machine-learning models that predict phytotoxicity across taxa based on easily measured physicochemical descriptors, accelerating safer-by-design formulation.

Formulation innovation itself represents a pivotal research frontier. Biodegradable polymer coatings and ligand-gated “smart” shells that respond to pH, root exudates, or moisture pulses can throttle nutrient release and minimize surges that trigger oxidative stress [103]. Next-generation platforms should integrate micronutrient cocktails or bio-stimulants within such coatings, creating multifunctional granules that sustain yield while reducing synthetic fertilizer loads. Coupling rigorous physicochemical profiling with multi-scale biological assays, as underway in our companion characterization project, will generate robust structure–activity relationships that regulators can translate into science-based application caps and labeling requirements. Collectively, these research priorities will convert laboratory insights into field-ready technologies, ensuring that nanofertilizers advance global food security without compromising environmental integrity [60,61].

## 5. Limitations and Biases

Despite certain methodological constraints, this research provides robust and reliable data that significantly advances our understanding of nanofertilizer phytotoxicity. The transparent acknowledgment of limitations demonstrates methodological rigor and scientific integrity while establishing critical safety benchmarks for agricultural nanotechnology applications [47].

### 5.1. Nanomaterial Characterization Constraints

The most significant constraint relates to the incomplete physicochemical characterization of tested nanomaterials. While particle size distribution was determined using dynamic light scattering (DLS) according to ISO 22412 standards, critical parameters including specific surface area, zeta potential, morphology, crystalline structure, and purity were not assessed [47]. These properties fundamentally influence nanomaterial biological activity and toxicity [104].

The study employs standardized particle size determination protocols ensuring reproducibility and comparability with international testing guidelines [47]. The observed dose–response relationships demonstrate internal consistency and biological plausibility, supporting the validity of phytotoxicity findings. Furthermore, ongoing comprehensive characterization studies using X-ray diffraction (XRD), scanning electron microscopy (SEM), and surface area analysis are actively addressing this limitation through a complementary Master’s thesis project [49].

### 5.2. Statistical Design and Sample Size Considerations

The experimental design employed duplicate replicates with 25 seeds per replicate (total N = 50 per treatment), which represents a conservative level of replication for comprehensive statistical analysis [105]. This constraint may limit statistical power to detect subtle treatment effects and increase uncertainty in parameter estimates. However, the sample size of 50 individuals per treatment exceeds the minimum requirements for EC_50_ calculations and variance estimation in phytotoxicity studies. The statistical design demonstrates methodological strengths, including appropriate randomization procedures, negative and positive controls, and standardized protocols that minimize systematic bias [105]. The replication level was constrained by practical considerations, including extensive physiological measurements per individual seed and occupational safety concerns when handling concentrated nanomaterials [106].

### 5.3. Biological Variability and Environmental Controls

Inherent biological variability in seed quality represents a potential source of experimental variation that was not systematically controlled [107]. Environmental variations during bioassay execution (temperature, humidity fluctuations) could contribute to observed variability between experimental runs [1,58]. However, the study implemented robust controls to minimize these sources of variability. *Lactuca sativa* seeds were obtained from a single commercial source and stored under standardized conditions to ensure batch consistency. Seeds were randomly distributed across the Petri dish surfaces, and experiments were conducted under controlled laboratory conditions. The consistent performance of negative controls (>90% germination) and appropriate response to reference toxicants validates the biological system’s reliability and confirms the absence of systematic bias.

### 5.4. Temporal and Exposure Protocol Limitations

The five-day bioassay duration, while standard for germination studies, may not capture delayed or chronic effects that could manifest over longer exposure periods [108]. The study focused on acute toxicity endpoints and did not assess potential recovery or adaptation responses with extended exposure. The five-day exposure protocol is well-established in phytotoxicity testing and consistent with internationally recognized guidelines for seed germination bioassays. This timeframe captures the critical germination and early seedling development phases most sensitive to environmental stressors and chemical toxicants [57,108]. The standardized protocol ensures comparability with existing literature and regulatory frameworks, enhancing the practical utility of findings.

### 5.5. Field Application and Regulatory Relevance

Laboratory-scale bioassay conditions may not adequately represent field exposure scenarios where nanomaterials interact with complex soil matrices, undergo environmental aging, and are subject to varying climatic conditions [68,108]. Despite these limitations, laboratory bioassays serve as essential screening tools that provide standardized, reproducible data for initial hazard assessment [53]. The controlled conditions allow for precise dose–response characterization that would be impossible under field conditions with multiple interacting variables. The EC_50_ values derived provide conservative safety benchmarks that can be adjusted using appropriate uncertainty factors for field applications [53]. The study successfully establishes clear dose–response relationships and derives statistically robust EC_50_ values that provide critical safety benchmarks for agricultural applications, representing a significant contribution to nanofertilizer risk assessment [49].

## 6. Conclusions

The comprehensive phytotoxicity assessment conducted in this laboratory-phase study provides critical foundational data for the safe development and implementation of nanofertilizer technologies in agricultural systems. Our findings demonstrate concentration-dependent toxicity patterns that reveal fundamental mechanistic insights essential for advancing sustainable agricultural nanotechnology. The observed dose–response relationships confirm threshold-mediated toxicity mechanisms rather than linear concentration dependencies, with critical transition points occurring between 0.5% and 1% concentrations for all tested nanomaterials. These findings align with reactive oxygen species (ROS)-mediated cellular damage pathways, where nanomaterial exposure triggers oxidative stress cascades that overwhelm cellular antioxidant defense systems [85]. The EC_50_ values of 2.13% for NF1, 22.8% for NF2, and 18.07% for NF3 establish quantitative safety benchmarks that demonstrate clear differential phytotoxicity among nanofertilizer formulations, with zinc oxide-based formulations (NF1) exhibiting the highest toxicity risk.

A particularly significant finding is the distinctive toxicity profile of clinoptilolite nanoparticles (NM3), which demonstrated superior safety margins compared with conventional metal oxide nanomaterials. The delayed toxicity manifestation and gradual dose–response relationship of clinoptilolite, attributed to its cation exchange properties and controlled ion release mechanisms, offers promising pathways for developing safer nanofertilizer formulations [86]. This finding supports risk-stratified approaches to nanofertilizer development, where materials with proven lower toxicity profiles should be prioritized for agricultural implementation while high-risk formulations require enhanced safety protocols. The laboratory-derived safety thresholds provide essential data for evidence-based regulatory frameworks, with environmental fate modeling suggesting that realistic field concentrations (0.1–0.3%) remain well below observed toxicity thresholds [53]. However, the potential for localized accumulation in rhizosphere environments emphasizes the critical need for species-specific application guidelines and comprehensive monitoring protocols.

These findings support the integration of nanofertilizers within comprehensive nutrient management programs rather than as standalone replacements for conventional fertilizers. The demonstrated safety margins for clinoptilolite-based formulations, combined with their controlled-release properties, offer particular promise for sustainable intensification in climate-vulnerable regions where precision nutrient management is essential for both productivity and environmental protection. The research framework employed here, as part of the broader FONTAGRO project (ATN/RF-18959-RG), demonstrates how laboratory screening studies can systematically inform field validation and practical agricultural implementation. The standardized bioassay approach employed in this study aligns with international testing guidelines and provides reproducible methodology for systematic nanofertilizer safety assessment.

Future research must address the identified limitations through comprehensive physicochemical characterization, long-term fate studies under realistic field conditions, and development of predictive models for species-specific sensitivity patterns [93]. The advancement of nanofertilizer science requires tiered research pipelines that progress from controlled laboratory conditions to crop-diverse, season-long field evaluations, supported by robust regulatory frameworks that balance innovation with environmental and human health protection. The implementation of nanofertilizers on commercial scales must be preceded by mandatory standardized testing protocols that integrate both efficacy and safety assessments from the earliest stages of material design. This precautionary approach is particularly crucial given the diverse compositions, behaviors, and interactions of nanomaterials in complex soil–plant–microbe systems.

This research demonstrates that while nanofertilizers present significant opportunities for enhancing agricultural productivity and sustainability, their successful implementation requires systematic risk assessment approaches that integrate mechanistic understanding with practical application guidelines. The differential toxicity profiles observed among nanomaterials support the development of safer-by-design formulations, while the standardized assessment methodology provides a foundation for regulatory decision-making. As agricultural systems face increasing pressures from climate change and food security demands, the responsible development of nanofertilizer technologies—guided by rigorous safety assessment and informed by mechanistic understanding—offers promising pathways toward sustainable agricultural intensification that balances productivity enhancement with environmental protection and human health considerations. The findings from this study contribute to establishing evidence-based guidelines that ensure nanofertilizer technologies advance global food security while maintaining environmental integrity and agricultural sustainability.

## Figures and Tables

**Figure 1 jox-15-00123-f001:**
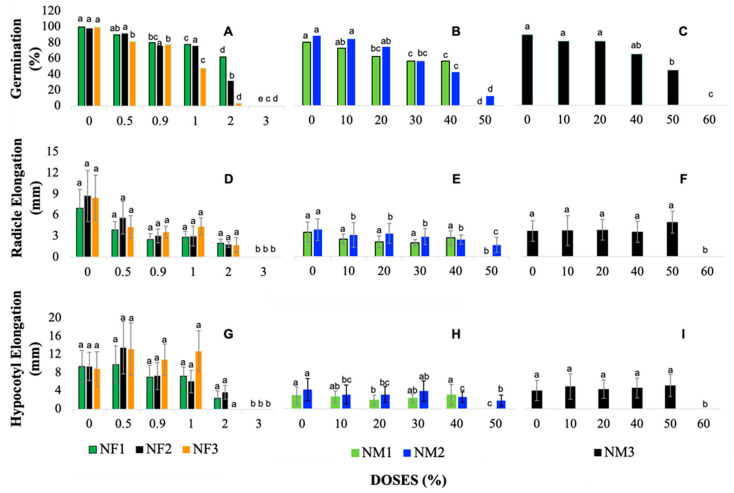
Dose–response patterns of key physiological parameters in *Lactuca sativa* (lettuce) seeds exposed to nanofertilizers (NFs) and nanomaterials (NMs): (**A**–**C**) Germination rates (%) for NFs, NM1, NM2, and NM3, respectively. (**D**–**F**) Radicle elongation (mm) for NFs, NM1, NM2, and NM3, respectively. (**G**–**I**) Hypocotyl elongation (mm) for NFs, NM1, NM2, and NM3, respectively. Different lowercase letters on the bars indicate statistically significant differences between doses (ANOVA, Tukey’s HSD, *p* < 0.05); full statistical results are available in the Appendix A (Appendix A).

**Figure 2 jox-15-00123-f002:**
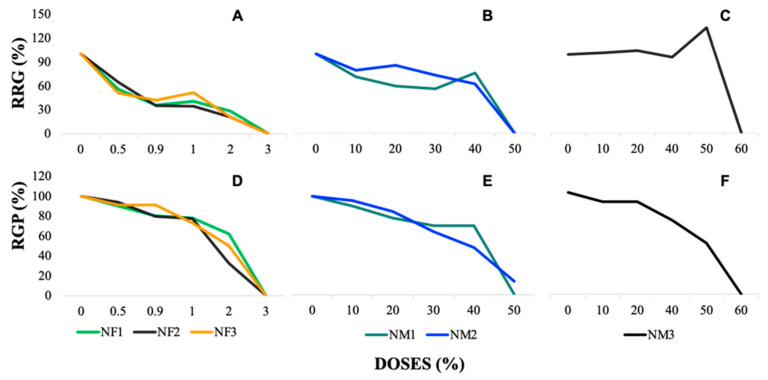
Relative radicle growth (RRG) and relative germination percentage (RGP) in *Lactuca sativa* (lettuce) seeds exposed to nanofertilizers (NFs) and nanomaterials (NMs): (**A**–**C**) RRG (%) for NFs, NM1, NM2, and NM3, respectively. (**D**–**F**) RGP (%) for NFs, NM1, NM2, and NM3, respectively. More details in Appendix A.

**Figure 3 jox-15-00123-f003:**
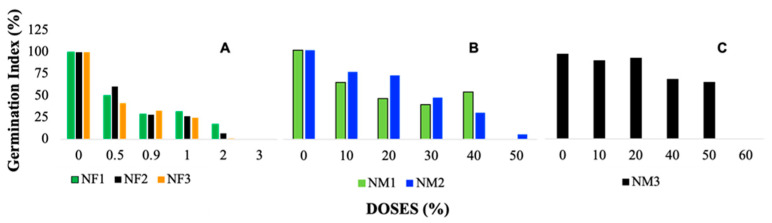
Germination index (GI, %) in *Lactuca sativa* (lettuce) seeds exposed to nanofertilizers (NFs) and nanomaterials (NMs): (**A**–**C**) GI (%) for NFs, NM1, NM2, and NM3, respectively.

**Table 1 jox-15-00123-t001:** Description of treatments evaluated.

#	Nanomaterials and Nanofertilizers	Abbreviation	Code
1	Zinc oxide	ZnO	NM1
2	Titanium dioxide	TiO_2_	NM2
3	Clinoptilolite	CLIP	NM3
4	Complete fertilizer * + Zinc oxide	Cf–ZnO	NF1
5	Complete fertilizer * + Titanium dioxide	Cf–TiO_2_	NF2
6	Complete fertilizer * + Clinoptilolite	Cf–CLIP	NF3

* Complete fertilizer: mixture of nitrogen (N) (13.30%), phosphorus (P_2_O_5_) (16%), potassium (K_2_O) (20%).

**Table 2 jox-15-00123-t002:** Ranking of germination index (GI%) in nanofertilizers.

Treatment	Dose (%)	GI (%)	Potential Toxic Effect
NF1	0	100	NE
	0.5	50	Moderate
	0.9	28	High
	1.0	32	
	2.0	17	
	3.0	0	
NF2	0	100	NE
	0.5	61	Moderate
	0.9	28	High
	1.0	27	
	2.0	7	
	3.0	0	
NF3	0	100	NE
	0.5	42	High
	0.9	33	
	1.0	25	
	2.0	1	
	3.0	0	

**Table 3 jox-15-00123-t003:** Ranking of germination index (GI%) in nanomaterials.

Treatment	Dose (%)	GI (%)	Potential Toxic Effect
NM1	0	100	NE
	10	64	Moderate
	20	46	High
	30	39	
	40	54	
	50	0	
NM2	0	100	NE
	10	76	Moderate
	20	72	
	30	47	High
	40	30	
	50	0	
NM3	0	100	NE
	10	92	Low
	20	95	
	40	70	Moderate
	50	67	
	60	0	High

**Table 4 jox-15-00123-t004:** Mean Effect Concentration (EC_50_) values obtained for the different nanomaterials and nanofertilizers under laboratory conditions.

Nano-Fertilizer	EC_50_ (%)	(Upper Limit–Lower Limit)
NF1	2.13	(1.06–3.44)
NF2	22.8	(17.12–30.52)
NF3	18.07	(15.12–22.26)
NM1	32.28 *	(30.23–34.56) *
NM2	30.03	(26.09–35.01)
NM3	33.57 *	(29.47–38.39) *

* Values marked with an asterisk (*) were estimated using the Moving Average method, which was applied only in the case of NM1 and NM3, where variability in control mortality made the Probit model unsuitable. The Moving Average method offers a descriptive EC_50_ estimate based on observed responses across doses, without assuming a specific distribution.

## Data Availability

The original contributions presented in this study are included in the article/Appendix A. Further inquiries can be directed to the corresponding authors.

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
