# Peer review of "Phytotoxic Effects and Agricultural Potential of Nanofertilizers: A Case Study Using Zeolite, Zinc Oxide, and Titanium Dioxide Under Controlled Conditions"

_jox, 2025, doi:10.3390/jox15040123_

Round 1
Reviewer 1 Report
Comments and Suggestions for Authors
Dear Authors,
Please find my observation for "Phytotoxic Effects and Agricultural Potential of Nanofertilizers: A Case Study Using Zeolite, Zinc Oxide, and Titanium Dioxide Under Controlled Condition"
- The introduction provides background on global population growth, greenhouse gas (GHG) emissions, and fertilizer use, but it does not clearly state the specific research gap considering the phytotoxic effects of nanofertilizers.
- In my opinion much of the introduction section content is generic that could be applied to many studies in sustainable agriculture. For example, statements like “Global population growth has intensified the demand for alternative approaches to achieve sustainable agricultural production” are broad and do not set up the unique context or focus of the manuscript - impact of nanofertilizers
- While Ecuador is mentioned as a case study, the introduction does not provide enough context about why Ecuador is particularly relevant or interesting for this research. There is a brief mention of Ecuador’s fertilizer use and vulnerability to climate change, but no specific details or statistics that would justify the focus on this country. For example the manuscript statements from L71-75 are widely applicable to many countries situation
- The research gaps considering the manuscript objectives are not well presented and clearly exemplified/sustained by currently available literature
- While particle size is measured using dynamic light scattering (DLS), other important characteristics (e.g., surface area, zeta potential, morphology, purity) are not reported. These properties can significantly affect biological outcomes and should be included for reproducibility and interpretation.
- The milling parameters are described, but there is no information on how the nanomaterials were stored after preparation, whether they were sterilized, or how potential contamination was avoided
- The description of preparation of nanomaterial suspensions for the bioassay is not complete in my opinion. For example, it is not specified whether the suspensions were sonicated, how homogeneity was ensured, or whether any dispersants were used. These factors can influence the effective dose delivered to seeds.
- The experimental design should clearly state how control and replication was considered. The section mentions a reference toxicant (potassium dichromate) and a control, but does not clearly describe the negative control (e.g., water only, or with dispersant if used). In my opinion the authors should be clear and explicit in presenting the role and composition
- For me is not enough clear how randomization in the placement of seeds or blinding in the measurement of outcomes were considered
- The text states that each treatment was performed “in duplicate,” (L218) which is a very low level of replication for statistical analysis. More replicates are generally required to ensure robust and reliable results up to general consideration
- The statistics section states that data were analyzed using one-way ANOVA and Tukey’s HSD, but does not mention whether assumptions (normality, homogeneity of variance) were checked. Please also mention how outliers or missing data were handled
- While the section mentions trends and differences (e.g., “a marked reduction,” “significantly reducing the germination index”), it does not provide specific statistical values such as p-values, confidence intervals, or effect sizes. For example, when stating that “there is a notable decrease in treatment NF1,” (L295) the actual statistical significance and the size of the effect are not reported
- The results often use vague language such as “notable decrease,” “marked reduction,” or “very low levels,” without quantifying these changes. For instance in L310 paragraph “germination starts near 90% and decreases abruptly” should be accompanied by exact percentages or means ± standard deviations to provide a clear view. This remark should be considered by authors through the entire manuscript where is applicable
- There is insufficient reporting of control group results. For example, the spontaneous mortality in controls is mentioned as exceeding a threshold, but the actual control values are not clearly provided
- In my opinion, there is insufficient reporting of control group results. For example, the spontaneous mortality in controls is mentioned as exceeding a threshold, but the actual control values are not provided. This is critical for interpreting the EC50 calculations and understanding the baseline from which effects are measured
- Please explain "Moving Average fit" what means or how it affects interpretation
- The discussion section primarily restates the results and aligns them with previous studies, but it does not provide a deep interpretation of the findings. For example, while it is mentioned that “the response of plants to nanoparticles may vary depending on the concentration, type of nanomaterial and plant species,” there is no exploration of the underlying mechanisms or biological explanations for these observations (L649-651)
- Several statements are generic and do not add specific insight. For instance, “Sustainable agriculture is a global challenge, and the implementation of nanotechnologies can be a valuable tool to improve efficiency and reduce environmental impact” is a broad claim that could apply to many studies. The discussion should focus on what is unique or novel about the current study’s findings (L682-684)
- The section does not critically evaluate the limitations of the study. For example, it does not discuss the low level of replication (duplicate samples), the high spontaneous mortality in controls, or any potential confounding factors such as variability in seed quality or environmental conditions during the experiment.
- While the discussion mentions the importance of phytotoxicity studies and the need for further research, it does not clearly connect the findings to practical implications for agriculture or policy. For example, how might these results inform the safe use of nanofertilizers in real-world settings? Are there recommendations for farmers or regulators?
- The section calls for “further studies” (L673) but does not specify what types of studies are needed. For instance, should future research focus on different plant species, long-term effects, field trials, or the environmental fate of nanomaterials? Specific recommendations would strengthen the discussion.
- Although some references to previous studies are made, the discussion does not thoroughly compare and contrast the current findings with the broader literature. The critical view should be seriously considered by authors
- The discussion mentions a project for corn production (L684), but the study itself was conducted on lettuce seeds. The relevance of the findings to corn or other crops is not explained, and this induce confusion
- The conclusion section should be reformulated considering the corrected version of the manuscript
Author Response
All answers, comments, and arguments are included in the attached file, "Answer to Reviewers 1 & 2."

Reviewer 2 Report
Comments and Suggestions for Authors
Dear Authors
The tittle: Phytotoxic Effects and Agricultural Potential of Nanofertilizers: A Case Study Using Zeolite, Zinc Oxide, and Titanium Dioxide Under Controlled Conditions.
The aim: ‘to determine the phytotoxicity of three types of nanomaterials (zinc oxide (ZnO), titanium dioxide (TiOâ‚‚), and clinoptilolite ((Ca, K, Na)₆(Si₃₀Al₆)O₇₂·20Hâ‚‚O) nanoparticles), three nanofertilizers, and potassium dichromate as reference toxicant (K2Cr2O7) using lettuce (Lactuca sativa) seeds as bioindicators under laboratory conditions, while also exploring the feasibility of their application in experimental agricultural production plots’.
- How do you measure the Agricultural Potential of Nanofertilizers? How is this part of the tittle represented in the aim? Is any measure you measured able to represent the Agricultural potential?
- How do you explore the feasibility of their application in experimental agricultural production plots? Do you mean a plot experiment? Or do you mean the cultivation in a farm? I understand, this is a next step and is not done in the manuscript. Do I guess properly, you are going to omit the pot experiment?
The material and method.
- Why did you take the Lactuca sativa? Why did you take the germination phase to reach the aim? This species is reach in cultivars. Please comment this fact.
- ‘The Lactuca sativa was used as a bioindicator, because its root system is significantly smaller compared to the aerial part’ How does it matter in case of seedling in cotyledons phase? I guess at this phase the experiment was terminated.
- ‘this species is highly sensitive to xenobiotic compounds’. Pease be much more specific. Yours discussion chapter is very short. Consider to shift the 2.3. subchapter text to the discussion and much more detailed explain.
- ‘33 adapting adequately to laboratory conditions’ text not clear something is missing.
- ‘with easily controllable growth factors’. ‘Easily’ seems not be a scientific argument. Please consider to write ‘well know methodology of cultivation under controlled conditions’ or/and Stuff possessing skills in this work, often used in many studies, a well-developed method
- ‘its germination occurs optimally between 18 and 25 °C’. How do the temperatures suit to the temperatures in which the germination process has been carrying on in the normal cultivation?
- ‘Another key component for its choice is that it has a fast germination process, allowing to observe toxic effects in a few hours of incubation’. Like 3.4 comment. Which of fifth parameters you exam requires to be ‘fast’? As an argument for me only decrease the manuscript value.
- What was the physiological stage of the seedlings when you stopped the growth? I guess cotyledons phase? What was the reasons for choosing this stage? In Agriculture the germination is the most crucial stage of the cultivation. So I have questions:
- Why did you chose such a critical stage?
- What is the representativeness of this plant stage to the rest of its vegetation?
- What was the seeds preparation process before sowing them? Was any pesticides used? How did you manage to avoid microbial infections? 100% of Germination Index did you reach sterile conditions?
- What was the time of the experiment versus the time of sowing lettuce in normal cultivation?
- Why didn’t you decide to have light at the end of the experiment? I mean when the cotyledons were raised above ‘the soil’
- Do you think, the seedling at so early (cotyledons)? stage has any possibility to respond for any fertilisers? Have possibility and need to uptake? Have time to give a response?
- How did you manage to weight, to measure the dry matter? How did you manage to remove NMs and NFs from the seedlings before weighting them?
- I can’t manage/understand the experiment with the potassium dichromate (Kâ‚‚Crâ‚‚O₇). Description is too short in subchapter 2.6. Was NM and NF used in this study? Where are the results? Discussion and conclusions? Why this xenobiotic was chosen?
- Is anything else (apart: seeds, water, NM, NF and Whatman No. 2 filter) added to the Petri dishes?
Introduction. Yours introduction is global, the human population, the world agriculture to feed them (me to so feed us). Global emissions form agriculture. Ok I accept. However:
- What are the resources of NM (zinc and titanium mostly) in compering with world’s agriculture or at least world lettuce cultivation?
- If in Ecuador 208 thousand tons of nitrogen is used a year what is the dose of nitrogen per a square unit of agricultural land? The numbers I count are very, very small. What threat to the environment such small dose may cause?
The Discussion. I do not accept the Discussion chapter. It should be written again.
- ‘This study is part of a binational FONTAGRO project between Ecuador and Colombia to develop nanofertilizer formulations for corn production, seeking to ensure that these productions are efficient, profitable and minimize greenhouse gas emissions. Please shift this sentence to Material and Method chapter.
- About the study you are reporting in this manuscript and stage you describe in the following sentences: ’Sustainable agriculture is a global challenge, and the implementation of nanotechnologies can be a valuable tool to improve efficiency and reduce environmental impact. However, it is essential to evaluate the safety and efficacy of these materials prior to their’ There is an enormous large gap. It is very good to ‘have a dream’ but writing dreams in the Discussion chapter is not yet acceptable!.
- What scientific questions or if you prefer scientific hypothesis did you asked or stated starting the study?? What were the hypothesis you verified in Anova? Write them down to improve the Discussion chapter.
- Did yours all results I mean the NM and NF amendments gave the lower (worse in some cases) results than control (water, seeds, paper, darkness, time)?. Why do you think is that? Try to explain the toxicity mechanism. What do you think NM and NF appeared to be toxic to lettuce seedlings?
- Please consider to join Results chapter with Discussion chapter.
- What are the concentrations of titanium an zinc on which the lettuce seedlings are being exposed? Please give the range and compere it with normal zinc concentrations is a soil.
- How are you going to overcome the toxicity of zinc and titanium?
- Consider to improve your discussion with manuscripts dedicated to zinc phytotoxicity studies.
Scientific soudnes and Intrest to Reader I marked first high, than I changed to low. And the same Final merit low. The potential is high however the way of presenting, the quality of the manuscript decrease the mark.
Author Response

(The authors gave the same response as above.)

Round 2
Reviewer 1 Report
Comments and Suggestions for Authors
Dear Authors,
Thank you for hard work in seriously considering my recommendation in the "Phytotoxic Effects and Agricultural Potential of Nanofertilizers: A Case Study Using Zeolite, Zinc Oxide, and Titanium Dioxide Under Controlled Conditions" manuscript. Reading carefully the new version of this manuscript I observed some issues that could be considered by authors. Please find them below:
- Supplementary material (page 10-11, NM treatments) - be consistent in presenting numbers "." vs ","
- L34: The authors should be accurate when names the applied procedure. If I know well the OECD-208 applies to terrestrial plant growth in soil. This study used Petri-dish germination on filter paper. If I have write, this strictly is an OECD-208-inspired germination assay, not the standard test, so the misleading should be corrected.
- Same about the number of cultivars recommended by OECD-208
- L374: Please provide justification for the selected dose "0, 0.5, 0.9, 1, 2, and 3%"
- For accuracy please write the formulas using "Equation" function of excel
Author Response
All answers and comments are in the attached file

Reviewer 2 Report
Comments and Suggestions for Authors
Dear Authors,
I accept the answers and improvements.
My advice to the Editor is accept the manuscript.
I wish you good luck.
Author Response
The response to Reviewer 2 is provided in the attached file
